# On the Calculation of Time Alignment Errors in Data Management Platforms for Distribution Grid Data [note 1]

**DOI:** 10.3390/s21206903

**Published:** 2021-10-18

**Authors:** Hans-Peter Schwefel, Imad Antonios, Lester Lipsky

**Affiliations:** 1Department of Electronic Systems, Aalborg University, 9220 Aalborg, Denmark; hps@es.aau.dk; 2GridData GmbH, 83278 Traunstein, Germany; 3Department of Computer Science, Southern Connecticut State University, New Haven, CT 06515, USA; 4Department of Computer Science and Engineering, University of Connecticut, Storrs, CT 06269, USA; lester.lipsky@uconn.edu

**Keywords:** electrical distribution grids, measurement errors, clock synchronization

## Abstract

The operation and planning of distribution grids require the joint processing of measurements from different grid locations. Since measurement devices in low- and medium-voltage grids lack precise clock synchronization, it is important for data management platforms of distribution system operators to be able to account for the impact of nonideal clocks on measurement data. This paper formally introduces a metric termed Additive Alignment Error to capture the impact of misaligned averaging intervals of electrical measurements. A trace-driven approach for retrieval of this metric would be computationally costly for measurement devices, and therefore, it requires an online estimation procedure in the data collection platform. To overcome the need of transmission of high-resolution measurement data, this paper proposes and assesses an extension of a Markov-modulated process to model electrical traces, from which a closed-form matrix analytic formula for the Additive Alignment Error is derived. A trace-driven assessment confirms the accuracy of the model-based approach. In addition, the paper describes practical settings where the model can be utilized in data management platforms with significant reductions in computational demands on measurement devices.

## 1. Introduction

Utilizing electrical measurements for grid operation and planning is common practice in higher voltage layers, and it is now also slowly being adopted in distribution grids. While the high-voltage grids deploy highly accurate and synchronized phasor measurement units [1], the operation and planning of medium- and low-voltage grids in practice need to rely on low-cost electrical measurements at transformer stations or junction boxes and increasingly on measurements from customer connection points, such as those provided by Smart Meters or by inverters that connect distributed energy resources. Such measurements of voltages, currents, and power are typically averaged over some logging interval of duration *T*, where *T* can range from a few seconds to tens of minutes [2]. It is important for grid monitoring and planning applications to account for the different sources of error in measurement data from such devices in their calculations. In addition to the consideration of measurement value errors caused by current transformers and the measurement device itself, the joint processing of electrical measurements from different measurement points in the distribution grid requires a quantification of the impact of imperfect clocks. An example of such applications is the calculation of power losses in a low-voltage grid area based on consumed and injected power at the substation and at all the customer connection points. While typical clock deviation errors of the participating measurement devices are in the order of few seconds [3,4], they can be larger in the case of infrequent clock synchronization or slow and highly variable communication delays on the communication network between master clock and measurement devices such as frequently observed in Smart Meter communication networks based on power-line communications or low-throughput Internet of Things communication technologies [5], which are highly dependent on wireless sensor networks (WSN). Reference [6] specifies an upper bound of ±20 s in synchronization error from a reference clock. The recognition of the importance of time synchronization support in WSN was the subject of recent work such as [7], where the authors propose three protocol-independent synchronization algorithms that observe the requirement of minimizing energy consumption. In [8], the authors discuss the introduction of new features in the updated IEEE 1588–2019 Precision Time Protocol (See [9]) that make it possible to overcome limitations of existing approaches.

To allow for a meaningful interpretation of imperfect measurement device clocks, earlier work [10,11] laid the foundations for investigating how the shift of measurement intervals affects the measured average values. As part of this earlier work, we introduced a Markov-modulated model that captures the behavior of power and voltage traces with a correlation structure that extends over long time periods. We further defined an approximation of the standard deviation of the time alignment error, denoted as ϵ. In [11] we carried out an initial study of the accuracy of the approximation based on measurement traces under limited analysis scenarios and a smaller set of measurement traces.

This paper builds on the earlier work to make this error quantifiable for online processing in electricity data management platforms. To that end, this paper

formally introduces a set of precise metrics to capture the behavior of the time alignment error (Section 3.3);shows that this measurement error is strongly dependent on the measured quantity and on time of day (Section 4.3 and Section 5.5), thus establishing that such error needs to be estimated online and cannot be replaced by a rule-of-thumb approximation;shows the challenges for measurement device complexity in a straightforward online estimation approach;introduces a model-based formula for Additive Alignment Error, assesses the accuracy of this model-based approach, and shows the benefits of applying the model-based online estimation in practical systems (Section 4.3).

## 2. Related Work

The quantification of the impact of measurement errors in different application contexts in general distributed systems received increasing attention in the last few years. As one example of relevant research, measurement errors at the sensor may propagate through the whole computation chain, see, e.g., [12] for work characterizing such errors and their impact. Specifically in energy grids, recent work addressed how to handle heterogeneous and noisy measurements in the context of grid estimation [13,14,15]. Those papers focus mainly on noise in the measurands and how to include such noise characterization in grid estimation procedures. In [16], the authors consider the state estimation problem in the presence of time synchronization errors, but their focus is on medium-voltage distribution systems where PMUs are available. In contrast to that, this paper addresses the impact of time alignment errors for measurands that are averaged over a time interval in low-voltage grids and are obtained from smart meters or other measurements devices with low time synchrony requirements. Distributed system state estimation in such grids is considered in [5]. The authors provide a methodology for analyzing the trade-off between accuracy and timeliness using on-demand measurements from smart meters without consideration of time alignment errors. A quantification of such errors, which is the goal of this paper, will make it possible to develop insights on the impact of time alignment error on grid estimation.

The impact of timing in access to measurement information in distributed systems was earlier analyzed in [17], and such analysis was put into context of different electricity grid applications and in generalized networked control applications. This paper instead focuses on the deviations of a measurand caused by time alignment deviations at the sensor.

The authors in [18] present an initial analysis of this aspect of the problem, however, their focus was on an empirical evaluation of the distribution of subsequent samples in a smart meter measurement trace. In contrast, this paper provides the definition of precise novel and useful metrics and it also presents a quantitative approach to calculate such error in data management platforms for distribution grid operators.

To capture autocorrelation of data over time, Markov modulated models were widely used. Even time series with long-range dependence, which are are common in many domains, can be successfully modeled by Markov models, see [19,20] for examples. The employment of a stochastic model, in addition to providing insights on time alignment error behavior under different setting, allows for a practical deployment without high computational burdens on measurement devices. This is discussed further in Section 5.

Earlier work [10] introduced a Markov-modulated model to study the impact of the timing error generated by the clock offset of the measurement interval, denoted as δ, on the value of the measured average. The model additionally provided an approximation of the standard deviation of the time alignment error, denoted as ϵ. While the previous paper provided a useful framework for analyzing this error, it was limited in its ability to capture the correlation structure of power and voltage traces, which extends over long time periods. To account for this correlation [11] introduced an extended model, referred to as MMP-GapT, along with a new approximation of the standard deviation of ϵ. It carried out an initial study of the accuracy of that approximation for a subset of the data used in this paper.

## 3. Quantification of the Time Alignment Error

The goal of this paper is to allow a Data Management Platform to provide relevant quantitative attributes on data quality together with the measurements to data analytics applications. Such data analytics applications can work on historic data, e.g., when executing loss calculation for a historic time period to identify inefficient low-voltage grids. In addition, there are near real-time executions of these data analytics applications, for instance, for anomaly detection and fault management. In both cases, historic or near real-time execution, such attributes on data quality are desired. This section defines the precise data quality metrics, starting off with previous work.

### 3.1. Summary of Previous Work

References [10,11] introduced and investigated the practically predominant measurement scenario in electricity grids, in which the true electrical variable is described by a continuous function m(t) and the measurement device determines the average value $\hat{m}\left(T,\delta \right)$ of m(t) over a time interval I=[δ,δ+T]:(1)$$\hat{m}\left(T,\delta \right)=1/T\int_{\delta }^{T+\delta }m\left(t\right)dt.$$

For convenience, the true measurement interval is placed to start at 0. The desired true average value is then achieved for δ=0, but a displacement δ>0 will be caused by nonideal clocks in the measurement device.

We consider m(t) as a stochastic process and define the random variable ϵ(T,δ) as the Absolute Measurement Deviation of the measured average value from its desired value. This Absolute Measurement Deviation results from the shift of the measurement interval by some displacement δ.
(2)$$\epsilon \left(T,\delta \right)=\hat{m}\left(T,0\right)-\hat{m}\left(T,\delta \right).$$

References [10,11] show examples of the distribution of this Absolute Measurement Deviation.

The resulting distributions of ϵ(T,δ) appear visually somewhat close to a zero-mean normal distribution. Reference [10] also proves that the expected value of ϵ(T,δ) is zero, when m(t) is characterized by a stationary Markov modulated process.

The averaging interval *T* is typically fixed by the measurement scenario, for instance motivated by voltage quality parameter definitions [2]. Electrical measurement deployments typically use T=10 min (See [10] for a first assessment of the impact of *T*). Note that this measurement interval duration is not so much determined by the measurement device itself, devices typically allow to measure values for intervals of 1 s or event down to the duration of a cycle of the AC voltage; however, the practically deployed devices aggregate over periods of several minutes to avoid an explosion of the data sizes for the communication network and for storage in a central headend server.

The value of δ depends on the clock synchronization protocol, the frequency of execution of clock synchronisation actions and on the quality of the internal clock in the measurement device. See [6] for an example of a specific clock synchronization approach.

### 3.2. Assumptions

We investigate methods for the data management platform to obtain the standard deviation of the Absolute Measurement Deviation for each measurement device and applicable to a defined time horizon (of, e.g., several hours). Practical assumptions for that are:The Data Management Platform can obtain an upper bound for the clock offset between the measurement device and a reference clock. This can be accomplished, for instance, by acting as time server for the measurement device using the Network Time Protocol [21];A constant offset between the measurement device clock and the reference clock is assumed for the period of interest, that is, clock drifts are low enough that the clock offset does not change in this period. This is a reasonable assumption if time periods of interest are in the range of hours to few days.

For the formal mathematical definitions and subsequent modeling, the stochastic process m(t) is also assumed to be stationary and ergodic over the time period of interest, and its expected value μ=E(m(t)) exists and is finite. We posit this is a reasonable assumption given that our analyses are based on partial traces (See Section 4.3).

### 3.3. Definition of Normalized and Additive Alignment Errors

We build on the previous work and focus on the standard deviation of the distribution of ϵ(T,δ). To make this measure independent of units, we normalize the standard deviation by the expected value of the measurand, μ, and call the resulting parameter Normalized Time Alignment Error:(3)$$\kappa \left(T,\delta \right)=\frac{std(\epsilon (T,\delta ))}{\left|\mu \right|}\cdot$$

Note that the normalization can optionally also be performed with respect to the upper range of the measurement device, as frequently performed for electrical measurement devices. Both normalizations lead to κ(T,δ) being unit-free.

To capture the Additive Alignment Error, we use the derivative:(4)$$\tilde{\alpha }\left(T,\delta \right)=\frac{d}{d\delta }\kappa \left(T,\delta \right)=\frac{1}{\left|\mu \right|}\cdot \frac{d}{d\delta }std\left(\epsilon \left(T,\delta \right)\right)$$

We are interested in the value of the Additive Alignment Error for small δ,
(5)$$\alpha \left(T\right)=\lim_{\delta \to 0}\tilde{\alpha }\left(T,\delta \right).$$

Therefore, the Additive Alignment Error tells us how much ’error’ is added by an increase of the disalignment of the time interval by each time unit.

## 4. Trace-Driven Assessment of the Additive Alignment Error in Customer Measurements

In this section, we perform a trace-driven assessment of empirical values of α(T) for a large set of measurements at customer connection points with one-second resolution. The purpose of this analysis is to show practically relevant ranges for the Additive Alignment Error, and to demonstrate that the online estimation is required because constant approximations are not likely to succeed.

### 4.1. Overview of Customer Measurements

The measurement data we use were collected by the ADRES project [22]. The entire data contains one-second measurements of phase-to-neutral voltages, active power and reactive power on each phase of the three-phase connections of 30 households in upper Austria taken during 2011. We use one week of data (604,800 samples each) for Phase A in summer and one week in winter. In total, this amounts to 180 one-week measurements (30 households, times 2 seasons, times 3 measurands) containing close to 109 million measurement samples.

Figure 1 shows the histogram of the voltage values of one selected one-week trace. The dotted vertical lines mark the boundaries that are used for discretization of the voltage values later for the modeling in Section 5.

### 4.2. Trace-Driven Calculation Method

To determine α(T), we select a fixed T=10 min to resemble a typical scenario of electrical measurements, motivated e.g., by voltage quality parameters [2]. For each of the 180 one-week traces, we calculate values of ϵ(T,δ) using a sliding window approach as follows: The one-week trace is divided into nonoverlapping measurement intervals, and for each interval a sample of ϵ is computed. The estimate of the standard deviation of the obtained values of ϵ is then used to calculate κ(T,δ). Computing κ(T,δ) over a range of values of δ and applying linear regression to them provides an estimate of α(T), since the latter is defined as the slope of κ(T,δ) as δ→0. The linear behavior of κ(T,δ) leads us to adopt the range δ=1 s, 2 s, 3 s…, 20 s in the linear regression calculation for the remainder of the paper.

Therefore, each of the 180 traces results in a single value of α(T). For presentation of the results, these 180 values are then grouped and processed as per measurand and per season. Later, in Section 5, the calculations of κ(T,δ) and α(T) from the traces are used as a basis for assessment of a modeling approach.

### 4.3. Trace-Driven Results for Additive Alignment Error

We analyze the traces to answer two questions: (1) what ranges of Additive Alignment Error result from the customer and grid behavior that is underlying the 1-second measurements? (2) Do we see a significant difference with respect to Additive Alignment Errors between different electrical measurands, seasons, and times of day?

Figure 2 provides the answer to both of these questions for the analyzed set of customer measurements. The α values computed are grouped by season, time of day, and electrical measurand, resulting in 18 clusters, each comprising 30 values of α. Energy consumption behavior is subject to daily profiles and weekly rhythms. For instance, weekends show different usage patterns than weekdays. Therefore, there is some inherent nonstationarity in grid-related measurements covering long time windows of more than a few hours. To investigate the effect of time of day, we carry out our analyses using extracts from the traces corresponding to the 2–5 am period from Monday through Friday, and the 1–4 pm period from Monday through Thursday (Friday is excluded as it can behave differently in Central Europe since shorter working days in many businesses on Friday are common). These are denoted as ‘morning’ and ‘afternoon’ in Figure 2. The averages, minima and maxima of these clusters are shown by the ‘X’ mark and the endpoints of the vertical lines in the figure. Among all the clusters, the morning traces produce the lowest average Additive Alignment Error, while the full trace produces the highest average, with the exception of the summer voltage measurements. Additionally, the minimum Additive Alignment Error obtained from the full trace is higher by one or two orders of magnitude than for the partial traces when looking at the power traces.

With regards to season, there is no strong impact on the additive error. The type of measurand, however, is significant. For active and reactive power, the average values of Additive Alignment Error are comparable, ranging from 0.3%/s to 1.1%/s, while the results for voltage traces show 2 to 3 orders of magnitude smaller averages. The observed behavior of voltages can be intuitively explained by the fact that voltages are a consequence of the behavior of all customers in the low-voltage grid.

Therefore, variability at a single customer does not influence voltage as strongly, while it directly influences active and reactive power of the measurement. It is important to note that the magnitude of the Additive Alignment Error associated with power measurements is comparable to measurement errors associated with smart meters, even when the clock deviations are significantly less than the ±20 s upper bound on error specified in [6] and despite the averaging period of T=10 min (which is a factor 30 to 600 larger than clock deviations of 1 to 20 s). Measurement errors range from ±0.1% to ±1.5%, depending on the smart meter’s class (see [23]). This indicates that an understanding of the overall error behavior must account for time alignment errors.

## 5. Model-Based Calculation of the Additive Alignment Error

The trace-based approach can in principle be implemented for online estimation of the Additive Alignment Error. Online estimation here means that this error is provided by a data management platform together with a batch of the measurement samples for the interval durations *T* after a period of assumed stationarity, e.g., normally during few hours. Figure 3 illustrates the context in which such a data management platform operates. However, such calculations of the Additive Alignment Error require high-resolution measurement data (e.g., 1-s), and since these high-resolution measurements for bandwidth and loading reasons are not transferred to the data management platform, such calculations can only be performed at the measurement device itself. On the other hand, the measurement device does not know about the precision of its clock. Therefore, each measurement device would be required to compute κ(T,δ) for a range of δ values using the sliding window approach. Calculating α would require the additional step of performing linear regression on the values of κ. The Data Management Platform then needs to receive the value of $\alpha_{i}$ for each measurement device *i*, or alternatively a set of values of values of κ(T,δ) for, e.g., δ is 1 to 20 s.

The approach shown in Figure 3 puts a high computational load on measurement devices, which would contradict the design goal in distribution grids to make these devices as simple and low-cost as possible.

The latter motivates to investigate a model-based approach as introduced and assessed further in this section.

### 5.1. Previous Work

Reference [10] used a continuous time Markov modulated process (MMP) to represent the measurand m(t) to obtain a first approximation of the standard deviation of the absolute measurement deviation, ϵ(δ,T). The MMP is fully characterized by its generator matrix *Q* and by a diagonal matrix, *E*, which contains the values of m(t), while the Markov Process is in the corresponding state; see [10].

However, the presented estimate and the underlying Markov modulated process suffer from severe drawbacks when applied to energy-related data, as shown in [11]: Such an MMP model does not account for the correlation structure inherent in electrical measurements, which frequently show significant auto correlation over longer time period.

In particular, the correlation at lag *T* is of interest, which can be seen by rewriting the definition of the Absolute Measurement Deviation as follows:(6)$$\begin{array}{ccc} \epsilon (T,\delta ) & = & \frac{1}{T}\left[\int_{0}^{T}m\left(t\right)dt-\int_{\delta }^{T+\delta }m\left(t\right)dt\right]\\ & = & \frac{1}{T}\left[\int_{0}^{\delta }m\left(t\right)dt-\int_{T}^{T+\delta }m\left(t\right)dt\right]\\ & = & \frac{1}{T}\left[L-R\right]. \end{array}$$

Therefore, the time alignment error is actually fully defined by the two intervals of size δ, whose integration provides the values *L* (left interval) and *R* (right interval).

The independence of the time alignment error from the behavior of the measurand in the time period [δ,T] was used in [11] to construct an improved model, referred to as MMP-GapT. The model uses the Markov modulated process from [10] only within the two intervals [0,δ] and [T,T+δ]. The matrices Q and E from the previous MMP model provide the corresponding behavior during these short intervals. They are both square matrices with dimension *N* given by the number of used states of the Markov model. An additional square matrix $P_{0,T}$ of same dimension is provided in addition to represent the state transition behavior between time 0 and time *T*; its elements are: $p_{i,j}=$ Pr(extended Markov model in state *j* at time T| extended Markov model in state *i* at time 0). The use of this matrix allows to represent correlation of the measurand on time scales of *T*.

### 5.2. Closed-Form Equation for Additive Alignment Error

When calculating ϵ(T,δ) from the left and right interval of size δ as introduced in Equation (Equation 6) in the previous subsection, the variance of ϵ(T,δ) then follows by standard rules on variance calculation:(7)$$Var\left(\epsilon \left(T,\delta \right)\right)=\frac{1}{T^{2}}\left[Var(L)+Var(R)-2\;COV(L,R)\right].$$

Our final target is the calculation of the additive error as defined in Equation (Equation 5). As this error is calculated for small δ, changes of the measurand m(t) during these intervals of size δ can be ignored, so the measurand takes value $m_{l}$ in the left interval and $m_{r}$ in the right interval. When using the MMP-GapT model to characterize the underlying measurand m(t), the probability distributions of $m_{l}$ and $m_{r}$ can be calculated:$$Pr\left(m_{l}=e_{i}\right)={\left(\pi \right)}_{i},\;Pr\left(m_{r}=e_{j}\right)={\left(\pi P\right)}_{j},$$
where $e_{i}$ is the unit vector with entry 1 at component *i*. We use $P=P_{0,T}$ as short notation above and from here on. The following expected values result:$$\mu_{l}=E\left(m_{l}\right)=\pi E\varepsilon^{\prime },\;\mu_{r}=E\left(m_{r}\right)=\pi PE\varepsilon^{\prime },$$
where $\varepsilon^{\prime }$ is a column vector with all *N* entries set to 1. Hence, the variance results in matrix notation (leaving out the details of the derivation):$$\begin{array}{c} Var\left(\epsilon \left(T,\delta \right)\right)=Var\left[\frac{\delta m_{l}-\delta m_{r}}{T}\right]\\ =\frac{\delta^{2}}{T^{2}}\pi \;\left[D_{l}^{2}+PD_{r}^{2}-2D_{l}PD_{R}\right]\varepsilon^{\prime }, \end{array}$$
where $D_{l}=E-\mu_{l}I$ and $D_{r}=E-\mu_{r}I$.

As a consequence, the Additive Alignment Error results for small δ:(8)$$\begin{array}{ccc} \tilde{\alpha }\left(\delta ,T\right) & = & \frac{d}{d\delta }\frac{std(\epsilon (T,\delta ))}{|\mu_{l}|} \end{array}$$(9)$$\begin{array}{ccc} & \;\;\;\;\;\;\;\;\;\;= & \frac{1}{T|\mu_{l}|}\sqrt{\pi \;\left(D_{l}^{2}+PD_{r}^{2}-2D_{l}PD_{R}\right)\varepsilon^{\prime }}. \end{array}$$

Note that the latter term is independent of δ, so it only depends on the matrix *P*, the diagonal matrix *E*, the steady-state probability of the Markov process π (which fulfills π·Q=0) and the aggregation interval duration *T*.

### 5.3. Model Fitting Choices

As shown in the last subsection, the MMP-GapT model allows for a closed-form calculation of the Additive Alignment Error. Such calculation can be performed by the Data Management Platform. The measurement devices need to only count the number of occurrences of the transitions from discretized values over time spacing *T*, and these counters can then be used by the Data Management platform to obtain the *P* matrix, see Section 5.6. Therefore, the Data Management Platform needs to first decide on the number of states, i.e., the dimensions of the *E* and *P* matrix.

To investigate the choice of the matrix dimensions, we apply the approach of selecting discretization bins of approximately equal number of measurement samples: the empirical cumulative distribution function (CDF) of the measurement trace is calculated and the value interval [0,1] of this empirical CDF is split into *N* equal bins. The empirical percentiles of the data are then used as the thresholds for discretization of the data. The vertical dotted lines in Figure 1 show an example of the resulting discretization thresholds for N=25 states. The *Q* and $P_{T}$ matrices of the model are then obtained from counting the one-step (*Q*) and lag-*T* transition probabilities within the discretized trace.

The solid blue curve in Figure 4 shows the resulting behavior of the calculated Additive Alignment Error from Equation (Equation 9) when fitting an MMP-GapT model with increasing number of states *N* to a specific trace containing a reactive power measurement.

The horizontal dotted blue line shows the value of α, which is obtained directly from the trace via the sliding window method described in Section 4.2. The convergence behavior for increasing *N* shows that a number of states N≥25 are required for the model to converge. This actually depends on the discretization approach adopted. Experiments based on more advanced ways to select the discretization thresholds show convergence for smaller *N*. These results will be presented in future work.

One reason for the slow convergence of the blue curve in Figure 4 lies in the fact that the used trace is covering a one-week measurement period. To investigate the effect of time of day, Figure 4 also show results with the green and cyan curves that are obtained from fitting to the morning and afternoon data sets introduced in Section 4.3, and the corresponding dotted curves show the value that result from the trace-based estimation of α. The green dashed line reflects the result from the afternoon trace, and it shows a much faster convergence behavior when fitting models of increasing *N* to the data, here approximately N≥14 is sufficient.

The cyan dashed curve uses only the morning trace. This shows a much lower variability of the data, leading to by a factor of more than 10 smaller α value, and convergence is achieved for rather small *N*. As another consequence of the small variability and of the in practice limited number of values that are provided in the measurement, only a limited number of different measurement values occur in the actual trace, making it impossible to fit models with larger N>17 to this partial morning data. For that reason, we always use N=15 in all assessments presented subsequently in this section.

### 5.4. Comparison of Models and Trace with Respect to κ(T,δ)

We seek to evaluate the accuracy of the MMP-GapT model, first with respect to an individual trace, and subsequently more generally across different traces.

For the first case, we consider the behavior of the Normalized Time Alignment Error κ(T,δ) introduced in Section 3 for three traces. We show results for T=600 and δ in the range 0 to 60 s to cover a sufficient range for the magnitude of the clock deviation that can be expected in practice. We also compare the resulting κ(T,δ) from the trace and the two simulations with a straight line approximation using the computed slope via Equation (Equation 9).

Figure 5 shows the results for three traces from different households of a reactive power measurement collected in winter. The figure shows how the traces compare to results from simulations of the MMP and MMP-GapT models. The choice of the three traces presented is illustrative of the range of results seen in the data set. Starting with Trace 47, the figure shows that both models and the straight line approximation provide a very good match for the result from the trace over the range of δ values. For Trace 48, the relative error grows more rapidly with increasing time offset δ, namely by 2.12% for each additional second of interval misalignment. In this case, the MMP model provides a slight overestimate, while MMP-GapT and the resulting computed linear approximation more closely match the result from the trace. Lastly, for Trace 46, which exhibits much fluctuation for δ>25, the MMP-GapT model provides a good match, while the MMP model simulations overestimate the relative error over the range of δ values.

As a final remark, the results in Figure 5 also demonstrate that the time alignment error is still relevant to be considered, even if technology evolution in practical deployments leads to much higher synchronization of clocks in measurement devices. Even if clocks are synchronized to an accuracy of 1 s or slightly below, the time alignment error visible in the results in Figure 5 is still in the order of measurement errors of superior-class metering devices, so this time alignment error is not becoming negligible, despite the quite long measurement period of T=10 min.

### 5.5. Comparison of Model Accuracy for α

Our goal is now to quantify the accuracy of the models and approximations with respect to the Additive Alignment Error. We use for comparison the relative deviation of the obtained value of α(T) for each trace *i*:(10)$$R_{i}=\frac{\left|\alpha_{i}^{\left(Model\right)}\left(T\right)-\alpha_{i}^{\left(Trace\right)}\left(T\right)\right|}{\alpha_{i}^{\left(Trace\right)}\left(T\right)}\cdot$$

For each trace *i* considered, we first calculate $\alpha_{i}^{\left(Trace\right)}\left(T\right)$ by the slope of the linear regression line of the empirical κ(T,δ) for δ in the range 1 to 20 s. Both the MMP and MMP-GapT models are then fitted to the trace and simulated to compute the same metric from the simulated traces from the model. Lastly, we compute the Additive Alignment Error from the matrix-algebraic formula in Equation (Equation 9) for the MMP-GapT model. The traces are aggregated by measurand, season, and time of day to investigate how significant these factors are with respect to model accuracy, resulting in six groupings of 30 traces each. For each grouping, we show the minimum, maximum, and average absolute deviation of the set of $R_{i}$ for the traces in the cluster. To investigate the effect of time of day on the accuracy on the models, we consider the same partial traces as the ones in Figure 2. These results are shown in Figure 6. The solid-color bars represent the mean of the $R_{i}$, and the lower and upper endpoints of the lines represent the minimum and maximum of the set of $R_{i}$, respectively. For each of the groupings, a small subset of traces (approximately 6%) is excluded from the results due to numerical problems in the data fitting procedure used to construct the matrices for the Markov model. However, this is unlikely to produce qualitatively different results; nevertheless, a closer examination of the fitting procedure will be left for future work.

The results for the morning traces, which are shown in Figure 6, indicate that on average, the MMP-GapT model simulations and the matrix algebraic computation offer a significant improvement over the MMP model across all trace groups. Additionally, the maximum deviations are lower for all groupings for both the MMP-GapT model and the matrix algebraic computation, with the matrix algebraic computation offering the lowest error rate for all groupings. To get a better understanding of the variability of α(T) for the matrix algebraic computation, we show the histogram of relative error of $\tilde{\alpha (T)}$ calculated by Equation (Equation 9) for the active power winter morning traces in Figure 7. The figure indicates that for the majority of traces, the relative errors $R_{i}$ are clustered in the range of 0 to 20%.

The results for the afternoon traces are shown in Figure 8. The same ordering among the analysis case holds for these trace as the morning traces with the exception of trace grouping for active power during summer (though caused by a single outlier). For all other groupings, the MMP-GapT model and the analytic approximation yield a significant improvement over the MMP model as before. While these results are qualitatively comparable to those of the morning traces when considering the mean relative absolute deviation, the magnitude of the maximum deviation are more pronounced, particularly for reactive power traces. This higher variability corresponds to the higher variability to the afternoon traces. A histogram of relative error, not shown here due to space limitations, indicates that for the majority of traces, the relative errors $R_{i}$ are clustered in the range of 0 to 30%.

In summary, these results indicate that the MMP-GapT model and the derived matrix algebraic computation are reasonably robust and accurate descriptors of the Additive Alignment Error under different measurement scenarios and measurands.

### 5.6. Practical Application of the Model-Based Approach for Online Estimation

We now summarize how the model-based approach for online estimation of the Additive Alignment Error is executed:The Data Platform sends discretization limits (and number of states) and time horizon for estimation (e.g., 3 h) to the measurement devices;The measurement device counts the transition matrix for discrete value changes (from which $P_{T}$ and also π can be obtained by the Data Management Platform) and sends it after the time period was elapsed;The Data Management Platform estimates the clock synchronization bounds, $\delta_{i}$, from measurements of communication network delays for each device *i*. If the Data Management Platform is also the time server for the clock-synchronization, then internal information from the clock-synchronization protocol execution can be used to improve this estimate;The data platform calculates $\alpha_{i}$ via Equation (Equation 7) and uses $\delta_{i}\cdot \alpha_{i}$ as the standard deviation of the clock induced error; this standard deviation is added to the standard deviation of the device-induced (and, if applicable, measurement transformer-induced) error and provided to the applications as data quality attribute.

The benefit of this model-based approach is that the measurement device only has to maintain a matrix of counters. Counting is a simple operation which can also be easily realized in hardware. All complex calculations are performed by the data management platform.

## 6. Summary and Outlook

The joint processing of electrical measurements from different measurement points in the grid requires a quantification of the impact of imperfect clocks. This paper formally introduces Additive Alignment Error, which captures the impact of misaligned averaging intervals of electrical measurements. The paper analyzes the behavior of this metric for a set of existing measurements at customer connection points in a low-voltage grid. The paper also introduces the MMP-GapT model as an extension of the Markov-modulated process from [10] and derives a closed-form matrix analytic formula for the Additive Alignment Error. A detailed assessment shows that the calculations from models fitted to the data result in a close match to the empirically obtained additive error. The benefits of the model-based approach regarding practical implementation of online estimation of this Additive Alignment Error using measurement devices with low computational capability are shown.

Although online estimation of the Additive Alignment Error for measurement data was the motivating use case, the presented model and analysis can also be useful for other purposes. One important question is how accurately clocks of measurement devices need to be synchronized, which then influences selection of communication solution and synchronization approach. The additive time alignment error results in this paper allow for a derivation of synchronization bounds from bounds of measurement errors. Therefore, the presented metrics and assessment approach is expected to become increasingly important for the future measurement solution design as it drives clock synchronization and communication architecture and protocol selection.

The inclusion of novel measurement devices with high synchronization (e.g., called micro-PMUs) in the distribution grid does not remove the need for such investigations as presented in this paper, as the deployment of micro-PMUs is limited, and hence the contribution of other data sources like AMI systems or inverter measurements remains high. Whenever including any of the latter measurement points, the assessment of the time alignment error, or inversely, the dimensioning of the clock synchronization approach, is required.

Future work will investigate the following directions: (1) Derive more robust fitting approaches for the MMP-GapT model using other discretization approaches for the measurement data; (2) analyze how to use the MMP-GapT model and online calculations of the Additive Alignment Error for quantification of total measurement errors in data management systems and observability applications, such as loss calculation [24], for distribution grids; (3) investigate a joint model of measurement errors and time alignment errors and its impact on data analytics applications in electrical distribution grids; (4) include consideration of incomplete measurement data in the online estimation procedures; (5) investigate the impact of the averaging period *T* on the time alignment error and on different data analytics applications (such as loss calculation), which utilize the measurement data. It is also expected that some theoretical limits can be obtained for a given δ/T for large *T*, and the practical application of such limit theorems may be investigated. 

References yes

## Figures and Tables

**Figure 1 sensors-21-06903-f001:**
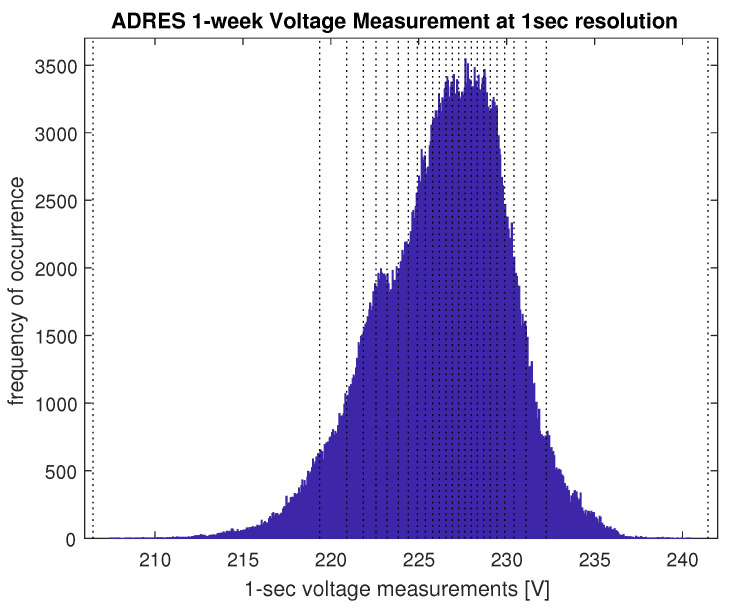
Histogram of one voltage measurements (including visualization of boundaries for discretization used later in Section 5).

**Figure 2 sensors-21-06903-f002:**
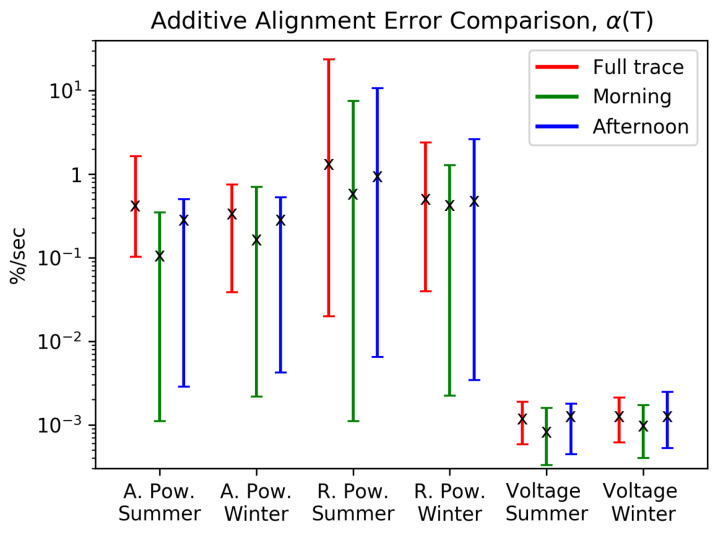
Additive Alignment Error α(T) for different electrical measurands at 30 customer connections obtained empirically from measurement traces for an averaging interval of T=10 min. ’X’ marks show average over values obtained from traces in each group (30 households, Phase A); endpoints of vertical lines show maximum and minimum value of α(T) observed in each group; note logarithmic scale on the y-axis.

**Figure 3 sensors-21-06903-f003:**
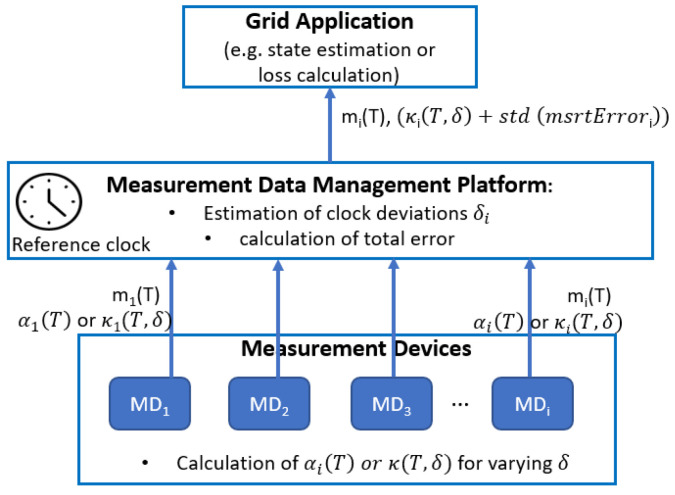
Calculation of time alignment errors by Data Management Platform of Distribution System Operator.

**Figure 4 sensors-21-06903-f004:**
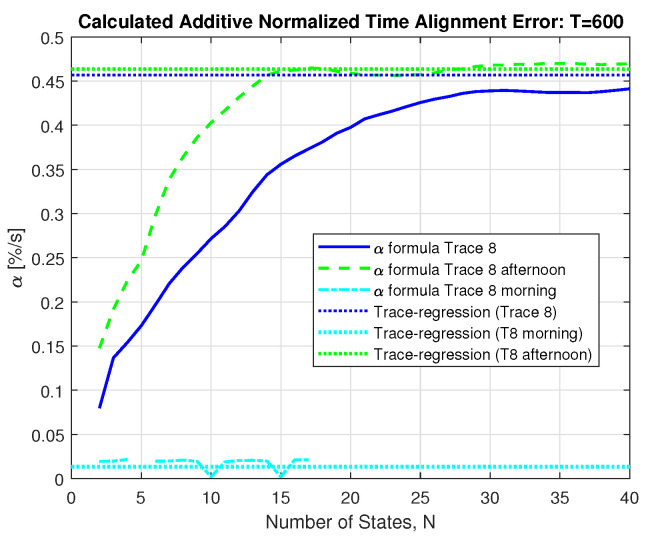
Resulting calculated α (T=600 s) when fitting a MMP-GapT model to Trace 8 for increasing number, *N*, of Markov model states—those discretize the data. Shown also are the results for the morning hours Monday–Friday 2 am–5 am, and for the afternoon hours Monday–Thursday 1–4 pm.

**Figure 5 sensors-21-06903-f005:**
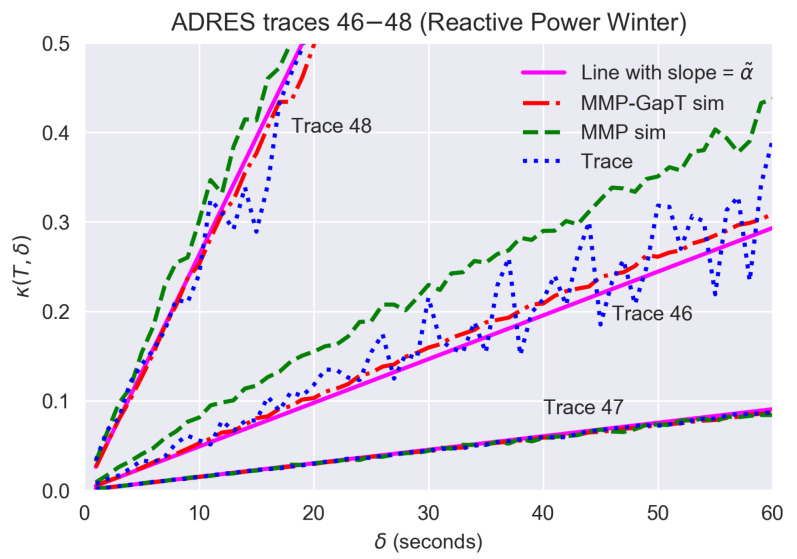
Comparison of Normalized Time Alignment Error κ(T,δ).

**Figure 6 sensors-21-06903-f006:**
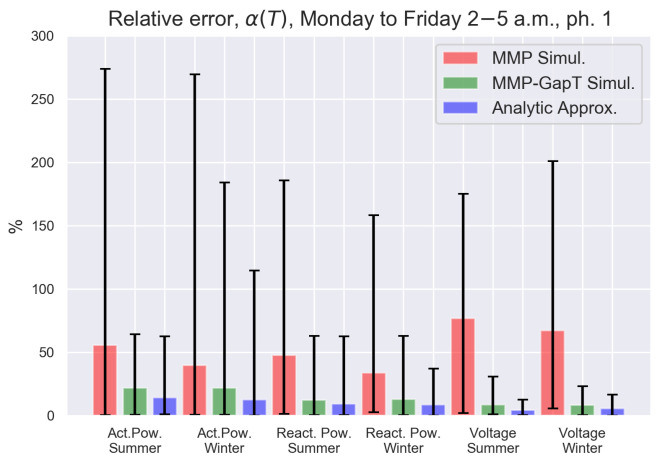
Comparison of relative deviation $R_{i}$ of Additive Alignment Error α(T) obtained from simulations of two models and from matrix analytic calculation of the MMP-GapT model relative to that from morning traces. Bars show the average for trace groupings, marked interval shows min and max of relative deviation of slopes.

**Figure 7 sensors-21-06903-f007:**
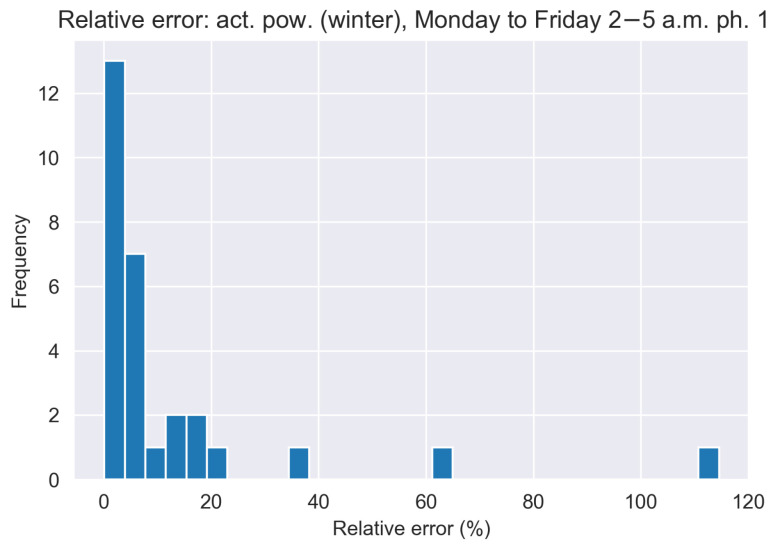
Histogram of relative error $R_{i}$ calculated from matrix analytic calculation based on MMP-GapT model for active power winter morning traces shown in Figure 6.

**Figure 8 sensors-21-06903-f008:**
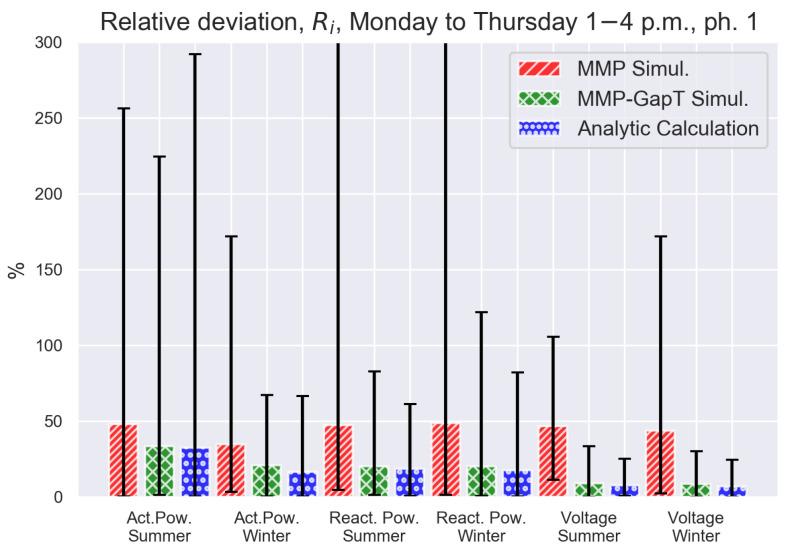
Same analysis as in Figure 6, but for afternoon traces. Note that vertical bars for the MMP simulations of reactive power summer and winter were clipped for improved presentation. Maximum values for these MMP simulation groups are 347% and 472%, respectively.

## Data Availability

Data sets used in this manuscript were obtained from ADRES-Concept: Autonomous Decentralised Renewable Energy Systems. These can be requested from https://www.ea.tuwien.ac.at/projects/adres_concept/EN/.

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
