# Peer review of "On the Calculation of Time Alignment Errors in Data Management Platforms for Distribution Grid Data†"

_sensors, 2021, doi:10.3390/s21206903_

Round 1

Reviewer 1 Report

The authors concern is current due to the importance of the accuracy of electrical measurements on data management platforms on electricity distribution networks. The present paper addresses the impact of time alignment errors for measurands that are averaged over a time interval. I appreciate the authors' efforts to capture the impact of misaligned averaging intervals of electrical measurements and solutions to minimize errors.

In my opinion, the references and arguments presented in the introduction section must also specify the Precision Time Protocols and Standards (e.g. IEEE 1588) and other work on the subject of the time alignment and synchronization (e.g. “Recent Advances in Precision Clock Synchronization Protocols for Power Grid Control Systems in Energies”, 2021, 14(17), 5303; https://doi.org/10.3390/en14175303).

Did you take into account the uncertainty of measuring the values used and then calculating the errors? It may be a future study.

Author Response

Thank you for the suggestion about the inclusion of references to the updated IEEE 1588 standard and other work on time alignment and synchronization.  We added references 7-9 and cited them at the end of the first paragraph of the introduction. 

Regarding the comment on measurement errors, we only focus on time alignment errors in this work.  An analysis of the joint effect of both types of error will be left for future work.  The last paragraph in Summary and Outlook provides some discussion.  

Reviewer 2 Report

The paper presents an interesting topic on the field of measurements for distribution grids. The proposed approach is interesting, however some aspects should be improved. Following some comments:

  • The present works seems an extension of the work published in [8]. It is recommended to better highlight in the introduction similarity and differences between the two works.
  • Related works sections should be extended with additional references face on the topic or consequences on possible applications (e.g. voltage control, state estimation, ...)
  • Sec. 4.2 you resemble the 1-sec data to 10 min. Why do you perform this? Averaging the values of the interval or downsampling?
  • Fig.3 delete the ")" at the end of the sentence
  •  

Author Response

Thank you for the thoughtful remarks.

We added some text at the end of the introduction to put the work in the context of earlier work as suggested by the reviewer.  This additional text is further elaborated on in the last paragraph of Section 2, and in sub-sections 3.1 and 5.1.  

We have added references [5] and [16] on state estimation to the Related Work section and put them in context of our paper.  

Regarding the comment about Section 4.2, the 10-minute interval is an averaging interval that is typical in such measurement scenarios to limit the communication and storage load in headend servers.  

We fixed the typo in the caption of Figure 3 and made some minor improvements to the presentation throughout the paper.  

Round 2

Reviewer 2 Report

The Authors answered my remarks appropriately. No additional comments, congratulation for the work.